# Polyphenol and Tannin Nutraceuticals and Their Metabolites: How the Human Gut Microbiota Influences Their Properties

**DOI:** 10.3390/biom12070875

**Published:** 2022-06-23

**Authors:** Marco Fabbrini, Federica D’Amico, Monica Barone, Gabriele Conti, Mariachiara Mengoli, Patrizia Brigidi, Silvia Turroni

**Affiliations:** 1Microbiomics Unit, Department of Medical and Surgical Sciences, University of Bologna, 40138 Bologna, Italy; m.fabbrini@unibo.it (M.F.); federica.damico8@unibo.it (F.D.); monica.barone@unibo.it (M.B.); gabriele.conti12@unibo.it (G.C.); mariachiara.mengoli2@unibo.it (M.M.); 2Unit of Microbiome Science and Biotechnology, Department of Pharmacy and Biotechnology, University of Bologna, 40126 Bologna, Italy; silvia.turroni@unibo.it

**Keywords:** polyphenol, nutraceutical, microbiota, metabolites, tannin

## Abstract

Nutraceuticals have been receiving increasing attention in the last few years due to their potential role as adjuvants against non-communicable chronic diseases (cardiovascular disease, diabetes, cancer, etc.). However, a limited number of studies have been performed to evaluate the bioavailability of such compounds, and it is generally reported that a substantial elevation of their plasma concentration can only be achieved when they are consumed at pharmacological levels. Even so, positive effects have been reported associated with an average dietary consumption of several nutraceutical classes, meaning that the primary compound might not be solely responsible for all the biological effects. The in vivo activities of such biomolecules might be carried out by metabolites derived from gut microbiota fermentative transformation. This review discusses the structure and properties of phenolic nutraceuticals (i.e., polyphenols and tannins) and the putative role of the human gut microbiota in influencing the beneficial effects of such compounds.

## 1. Introduction

Nutraceuticals are generally defined as “a food or part of a food that provides benefits to health in addition to its nutritional content” [1]. In this sense, prebiotics and plant-derived molecules can certainly be included in such a definition.

By “phenolic compounds”, we generally refer to ubiquitously distributed plant-derived phytochemicals, which are part of our daily diet due to their presence in both fruit and vegetables. They are generally synthetized as secondary metabolites via the shikimic acid and phenylpropanoid pathways, mainly to provide protection against potentially harmful phytopathogens [2]. The two main classes of phenolic compounds are polyphenols and tannins. Polyphenols are generally low-molecular-weight compounds that can be divided into multiple classes based on their structures, including flavonoids, stilbenes, phenolic acids, lignans, non-phenolic metabolites and other polyphenols [3]. In particular, flavonoids can in turn be divided mostly into flavonols, flavanones, isoflavones, flavones, flavan-3-ols and anthocyanins, while phenolic acids can be principally represented in the two simplest scaffold structures of hydroxybenzoic and hydroxycinnamic acids [4]. Each of these families and sub-families hosts numerous structural variants sharing the same family-defining scaffold. Indeed, around 8000 different polyphenolic compounds have been identified in plants to date, and this number is expected to increase in the coming years [5]. On the other hand, tannins are a class of phenolic biomolecules with a high molecular weight, ranging from 500 Da to 20 kDa and can be mainly classified into the following three classes: hydrolyzable tannins, condensed tannins and phlorotannins—based on the stability or origin of their structure [6].

Both polyphenols and tannins have shown discrepancies in the explanation of their positive effects in vivo when assessing the effective concentration levels in plasma, raising doubts about their bioavailability, which is generally very low [7,8]. In this regard, the food matrix has been shown to exert a relevant impact on nutraceutical bioavailability [9], and even when evaluating the effects of phenolic compounds-rich extracts, it should be remembered that multiple nutraceuticals are present at the same time, thus raising difficulties in determining the molecules—most probably metabolites—involved in providing the biological effects. It is very likely that the original phenolic molecule is not the one absorbed in the bloodstream and linked to the effects on the human body. In this scenario, more and more studies have suggested and sometimes tested the contribution of the gut microbiota in metabolizing dietary phenols into active biomolecules, which are actually responsible for the health benefits [10,11,12].

For the purpose of this review, we focused our attention on the following two broad classes of plant nutraceutical compounds: polyphenols and tannins. Specifically, given the vast structural diversity of polyphenols, we focused on flavonoids, stilbenes, and phenolic acids. For each nutraceutical class, we first aim to clarify the subgroup partitioning. For each subgroup, we then discuss their main properties, structure and the role of the gut microbiota in influencing their bioavailability and the generation of bioactive metabolites, affecting, therefore, the beneficial effects on human physiology. All articles presented in this review were identified using the PubMed platform, the full-text archive of biomedical and life sciences journal literature at the United States National Institutes of Health’s National Library of Medicine. The most appropriate and relevant for each topic were selected and commented on.

## 2. Polyphenols and Tannins

In the following sections, we describe the several selected families and sub-families of polyphenols, and then conclude the overview outlining tannins. For each class of compounds, we first summarize the main food sources, then discuss the known in vitro and in vivo properties, followed by a description of the best-known structural variants. Finally, we consider the role of the gut microbiota in the metabolism of the considered compounds and in the generation of known bioactive molecules. A summary of all the in vivo and in vitro studies considered in this review can be found in Appendix A.

### 2.1. Flavonoids

#### 2.1.1. Flavonols

Flavonols are the most widespread form of flavonoids in plant foods, especially in onions (*Allium cepa*), strawberries (*Fragaria* spp.), spinaches (*Spinacia oleracea*), blueberries (*Vaccinium* sect. *Cyanococcus*), cauliflowers and broccoli (*Brassica oleracea*). The best-known compounds in this sub-family are quercetin and kaempferol.

In vitro, quercetin has shown anticancer, apoptosis-inducing [13] and antioxidant activities [14]. Similarly, kaempferol has been associated with antioxidant, anti-inflammatory, anticancer and antimicrobial effects [15]. In vivo, several clinical trials have been conducted, finding general associations between flavonols intake and a reduction in cardiovascular risk factors [16,17]. Among the biochemical properties of flavonols, a marked antibacterial activity has been reported [18], and they have also recently been proposed as adjuvants to antiviral drugs, acting against SARS-CoV-2 [19] and other viral pathogens in general [20].

Their structure is characterized by a 3-hydroxyflavone backbone, with the phenolic hydroxyl moieties being responsible for the diversity of this class of polyphenols (Appendix A). They are mostly found in glycosylated forms in foods, predominantly complexed with glucose or rhamnose, and the type of glycosylation affects their metabolism as follows: the higher the number of saccharides in the complex, the slower the metabolism of the compound (especially for the trisaccharide forms) [21]. Usually, the glycosylated forms are found as O-glycosides, with substitution occurring at many sites of the flavonol backbone. Besides the aglycones quercetin and kaempferol, myricetin is also a widespread flavonol, together with the methylated glycosidic derivative isorhamnetin. When conjugated with saccharides, quercetin and kaempferol glycosides possess hundreds of different glycosidic combinations, dramatically expanding the diversity of flavonols that can be found in food. Notably, rutin is one of the main sources of flavonols in food [22] and constitutes a complex glycoconjugate form of quercetin.

The breakdown of flavonols begins in the oral cavity, with the saliva and oral microbiota initiating the conversion of the flavonol glycosides to their aglycone form [23], a process that proceeds in the upper digestive tract. Here, glycosidic forms are poorly absorbed and, in the large intestine, they undergo most of the metabolism through α-rhamnosidase and other glycosidase activities provided by the gut microbiota. In particular, exploiting fluorescence-based single-cell activity measurement coupled with fluorescent activated cell sorting after anaerobic incubation of healthy human fecal microbiota with rutin, Riva ang colleagues [24] detected an enrichment in the bacterial families *Lachnospiraceae* (specifically, the genera *Lachnoclostridium* and *Eisenbergiella*), *Enterobacteriaceae (Escherichia)*, *Tannarellaceae* (*Parabacteroides*) and *Erysipelotricaceae* (*Erysipelatoclostridium*), suggesting that they carry the metabolic capacity to degrade rutin. Other species such as *Lactobacillus acidophilus*, *Lactobacillus plantarum* and *Bifidobacterium dentium* were reported to have a high rutin-deglycosylation capacity in in vitro fermentation studies, releasing the sugar moiety and the aglycone quercetin [25,26]. Flavonol aglycones can be partially absorbed by epithelial cells in the upper and lower intestine and undergo phase II metabolism with the production of O-glucuronide and O-sulfate conjugates that can be found in plasma and urine [27]. However, bioavailability studies recorded only a small amount of the flavonol structure in plasma, mainly because the highest fraction of the aglycone is not absorbed through the epithelium but rather undergoes metabolic processing by the gut microbiota. Multiple components of the intestinal microbial ecosystem, such as the species *Bacteroides fragilis*, *Clostridium perfringens*, *Eubacterium ramulus*, *Lactobacillus* spp., *Bifidobacterium* spp. and *Bacteroides* spp., showed in mouse models the potential to degrade aglycones, releasing ring-fission phenolic acid end-products, including 3(3,4-dihydroxyphenyl) acetic acid, 3,3-hydroxyphenylpropionic acid, 3,4-dihydroxybenzoic acid and 4-hydroxybenzoic acid [28], some of which are known to exert free-radical scavenging and antioxidant activity [29].

#### 2.1.2. Flavanones

Flavanones, generally represented by hesperetin and naringenin, are widely distributed in the plant families *Compositae* (as lettuce, chicory, artichoke), *Leguminosae* (such as chickpeas, peanuts, lentils, peas, beans) and *Rutaceae* (such as cedar, tangerine and lemon, belonging to the *Citrus* spp. fruits, which are the main source of flavanone glycosides, mainly hesperidin) [30]. Flavanones possess antioxidant and anti-inflammatory activities [31] and constitute important scaffolds for the development of anti-inflammatory and anticancer therapeutic agents [32]. Several clinical trials have tested the in vivo effects of flavanone intake, mainly focusing on their ability to exert protective effects on the cardiac function, but providing contradictory results [33], or on their ability to improve endothelial function [34]. Concerning epithelial barrier function, *Citrus* extract has been shown to enhance β-carotene uptake in intestinal Caco-2 cells by inducing paracellular permeability and directly interacting with the membrane [35], thus suggesting that the pleiotropic mechanism of action of flavanones could begin directly in the intestinal lumen.

Flavanones generally show antibacterial potential [36], with those extracted from *Calcedaria thyrsiflora* (a succulent plant commonly called “kalanchoe”) exhibiting inhibitory activity against methicillin-resistant *Staphylococcus aureus* (MRSA) [37]. They also possess antiviral activities and have recently been tested to possibly aid in anti-COVID-19 therapy [38].

This polyphenol subfamily exists in both glycosidic and aglyconic forms (Appendix A). Hesperetin and naringenin are the best-known aglycone forms, whilst the glycoside counterparts hesperidin and naringin are bound to the disaccharide rutinose and rhamnose-β1,2-glucose, respectively. The deglycosylation process—starting from saliva and continuing in the upper digestive tract—is likely analogous to that of other flavonoids and generally shared among all the glycosylated biomolecules ingested. Once released from the glycoside moiety, the aglycone naringenin has been tested for fermentation by the gut microbiota in rats [39] and its metabolism yielded a high number of metabolites, mainly phenolic acids, *p*-hydroxyphenylacetic acid and 3-(*p*-hydroxyphenyl) propionic acid. It has recently been confirmed that also the flavanones hesperidin and eriocitrin (a glycosylated flavanone extracted from lemons, e.g., *Citrus limon*) are first deglycosylated in the upper digestive tract, then metabolized in the lower intestine with a complex series of metabolic transformations and interconversions, involving methylation, followed by glucuronic acid and/or sulfonate conjugation [40]. Phase II flavanone metabolites naringenin 7-O-glucuronide, hesperetin 3′-O-glucuronide and hesperetin 7-O-glucuronide, together with the microbiota-derived flavanone metabolites hippuric acid and 3-(4-hydroxyphenyl) propionic acid showed anti-inflammatory and antioxidant activities in vitro at physiologically tangible concentration [41]. Several bacterial species have been associated with the core metabolic steps required to yield flavanone metabolic end-products, such as O-deglycosylation (among which *Parabacteroides distasonis*, *Bifidobacterium adolescentis*, *Bifidobacterium bifidum*, *L. plantarum*, *Lactobacillus buchneri*), C-deglycosylation (*Eubacterium cellulosolvens*), O-demethylation (*Eubacterium limosum* and *Blautia* sp. MRG-PMF1), dihydroxylation or general C-ring cleavage (*Clostridium butyricum*, *E. ramulus*, *Flavonifractor plautii*) [42].

#### 2.1.3. Isoflavones

Isoflavones are generally found in soybeans (*Glycine max*, the richest source of these biomolecules), chickpeas (*Cicer arietinum*), pistachios (*Pistacia vera*), peanuts (*Arachis hypogaea*) and other nuts and legumes. Isoflavones are often found as glycosides, most commonly daidzin, genistin and glycitin, with daidzein, genistein and glycitein being the corresponding aglycones (Appendix A). As a common feature of polyphenols, isoflavones exhibited in vitro antioxidant, antimicrobial, anti-inflammatory and anticancer properties as well [43]. It is well known that isoflavones act as phytoestrogens by binding to estrogen receptors in mammals [44], and this has raised several concerns given the potential endocrine-disrupting effect. However, a recent critical review [45], which considered 417 reports on soybean-derived products and isoflavone phytoestrogen effects on humans, concluded that they do not act as endocrine disruptors, but rather are beneficial for their antioxidant and anti-inflammatory effects. Furthermore, the coupled administration of isoflavones and probiotics has recently shown long-term efficacy in counteracting the loss of bone mineral density in a double-blind, placebo-controlled trial in postmenopausal osteopenic women [46].

Regarding the antibacterial activity, isoflavones have shown inhibitory effects against biofilms of *Escherichia coli* and *Listeria monocytogenes* [47], as well as against MRSA [48]. Moreover, ingestion of isoflavones has been shown to potentially reduce viral infection and mortality of porcine reproductive and respiratory syndrome in pigs, with evident outcomes relevant to the food supply chain [49].

In vivo, isoflavones are mainly ingested as glycosides, so a high fraction of their content reaches the large intestine, where they are metabolized by the gut microbiota. Conversely, aglycone isoflavones are absorbed by small intestinal endothelial cells and metabolized through phase II reactions, yielding glucuronide and sulfate metabolites, as demonstrated by in vitro experiments [50]. These conjugates can be excreted in the bile, re-enter the intestine and, here, be converted back into aglycone forms by enzymes of microbial commensals, such as β-glucuronidases. The resulting compounds, mainly dihydrodaidzein and dihydrogenistein, can again be further metabolized by the colonic microbiota. It appears that the configuration of the intestinal microbial community plays a major role in determining the end-product that will be generated, alternatively equol or O-desmethylangolensin (O-DMA) [51], with *Clostridium* spp. And *E. ramulus* being among the main bacteria involved in such decision-making balance [52]. It should be noted that the equol-producer phenotype is less prevalent in the population than the O-DMA-producer one and, given that equol shows better binding properties to α and β estrogen receptors than O-DMA and unprocessed isoflavones, this might result in different physiological effects and health benefits [53]. For what concerns the effects of equol and O-DMA on the human body, the former showed potential anti-atherogenic effects, possibly preventing stroke [54], while for the latter, only a few studies have evaluated disease risk factors in relation to being an O-DMA producer, but apparently the O-DMA producer phenotype might be associated with obesity in adults [55]. This duality kindles the interest in deepening our knowledge of the relationship between the gut microbiota and isoflavones and polyphenols in general—given their widespread presence in our daily dietary foods.

#### 2.1.4. Flavones

Flavones are a subfamily of flavonoids commonly reported in parsley (*Petroselinum crispum*) and celery (*Apium graveolens*), but also present in grains including maize (*Zea mays*), wheat (*Triticum* spp.), rye (*Secale cereale*), barley (*Hordeum vulgare*), oats (*Avena sativa*), sorghum (*Sorghum* spp.) and millet (*Pennisetum glaucum*). The vast majority of flavones reported come from cereal grains and exist as various O- or C-glycosides of the aglycones apigenin and luteolin, forming a huge variety of structural variants. In fact, the different glycosylations can generate several glycosidic forms depending on the following: (i) the number of saccharide units; (ii) the type of glycosyl moiety; (iii) the position in which the saccharides bind to the flavone backbone (Appendix A). For example, the saccharides bound to position 7 of the A-ring of apigenin can generate apigetrin (1 unit of glucose), apiin (a disaccharide of furanose and glucose) or rhoifolin (a disaccharide of rhamnose and glucose). Most of the in vitro studies have used aglycones or unprocessed glycosides, yielding controversial results, probably due to the lack of a specific metabolism to make the flavone configuration actually functional. Conversely, results with synthetic derivates of flavones showed consistent and strong effects, mainly associated with antioxidant activity and inhibition of the lipoxygenase enzyme [56]. Anticancer activity often associated with several flavonoids has also been reported for apigenin and luteolin, inducing apoptosis in tumor cells, but the limited in vivo evidence requires further investigation [57,58]. Finally, flavones have generally been associated with antiviral, antibacterial and antifungal effects [20].

As mentioned for the other flavonoid subfamilies, ingested flavones are mainly glycosides and, in such form, they reach the intestine, with partial degradation in the oral cavity and in the upper digestive tract. The aglycone form can be absorbed and found in plasma together with glucuronide and sulfate forms [59]. An interesting effect is reported for baicalin, a glucuronide form of baicalein—a flavone mainly derived from the roots of *Scutellaria baicalensis*—that shows enhanced absorptive and bioactive potential after undergoing glucuronidation [60]. Both the flavone forms that reach the intestine directly and those that return to the intestine after hepatic metabolism can be further metabolized by the gut microbiota. In particular, flavone O-deglycosylation appears to be widely distributed among gut microbes, in genera such as *Eubacterium* [61], *Flavonifractor* and *Clostridium* [62]. The released aglycones are mostly further fermented by colonic bacteria with C-ring cleavage, releasing 3-(3,4-dihydroxyphenyl) propionic acid, 3-(4-hydroxyphenyl) propionic acid, 3-(3-hydroxyphenyl) propionic acid and 4-hydroxycinnamic acid. Such phenolic acids are absorbed and circulated in the bloodstream until their excretion in the urine [63]. In particular, for 3-(4-hydroxyphenyl) propionic acid, Xie et al. [64] reported associations with a reduction in total cholesterol in healthy adults.

On the other hand, it must be said that the introduction of phenolic molecules through the diet is likely to exert an impact on the gut microbiota itself. Tangeretin and nobiletin are other flavones isolated from the tangerine peel (*Citrus reticulata*), which are an example of polymethoxyflavones, a subclass of flavones bearing two or more methoxy groups on the basic benzopyrone. Such biomolecules have been shown to positively alter the composition of the mouse gut microbiota after ingestion, with an increase in the relative abundances of the genera *Lactobacillus* and *Bifidobacterium* [65]. These taxa are in turn involved in polymethoxyflavone metabolism, thus paving the way for studies focused on the bidirectional relationship between polyphenols and gut microbes.

#### 2.1.5. Flavan-3-ols

Flavan-3-ols is a collective term for several types of so-called catechins, which are mainly found in apples (*Malus* spp.), hops (*Humulus lupulus*), tea and black tea (*Camellia sinensis*), beer, wine, fruit juices and cranberries (*Vaccinium* spp.). *In vitro*, flavan-3-ols have been shown to inhibit tumor angiogenesis and TNF-α-related inflammation [66] and have recently been tested in association with procyanidin B2—an anthocyanin flavonoid—in breast and prostate cancer cell lines, finding out that the combined treatment efficiently enhanced a sensitization mechanism that could be exploited in novel clinical trials [67]. Sensitization of cancer cells often requires a combination of therapies with multiple drugs and aims to achieve a synergistic induction of cell death in cancer cells. When tumor cells are resistant to therapy, the combination of drugs can enhance the antitumoral activity by modulating one or more mechanisms of resistance. Among the potential chemosensitizers in use, 60% of them are of natural origin [68,69]. Further evidence of the cancer-protective effects of flavan-3-ols has been reported in a meta-analysis on various cancers, including rectal, oropharyngeal, laryngeal, breast and stomach cancers [70]. Flavan-3-ols have also shown antioxidant properties, mainly when extracted from natural food matrices [71]. In case-control studies, the intake of foods rich in flavan-3-ols—mainly tea extract and green tea infusions—was associated with a lower risk of type-2 diabetes [72] and cardiovascular diseases [73]. These results, far from being exhaustive, should encourage further research on flavan-3-ols and polyphenols in general as promising molecules.

Flavan-3-ols and, in particular, foods rich in such biomolecules, have also been shown to be effective in counteracting viral infections [74]. Catechins and derivates exhibited antibacterial properties as well, on both Gram-positive and Gram-negative bacteria. The synergistic effects of antibiotic treatments have also been reported in clinical trials [75], paving the way for future studies to overcome antibiotic resistance, by reducing antibiotic dosage while maintaining and possibly increasing therapeutic efficacy (with alleviation of the economic burden on healthcare systems).

The flavan-3-ol structure accounts for the ease of free radical scavenging activity, primarily due to the high reactivity of hydroxyl substituents in the flavan-3-ol backbone (Appendix A). The best-known flavan-3-ol compounds are catechin, epicatechin, gallocatechin and epigallocatechin, as well as their galloylated forms. In addition to the possible substituent, the catechin’s potential structural diversity relies on two chirality centers. The epicatechin scaffold carries both chirality centers pointing in the same direction (*cis*), whereas that of catechin shows opposite directions (*trans*). Galloylation—together with the multiple hydroxyl groups—results in the formation of a hydration shell that determines reduced bioavailability and absorption in the upper digestive tract, thus making flavan-3-ols reliant on gut microbial metabolism—attributed to several *Lactobacillus* spp. [76]—to break down the ester bond and release the catechin flavonic backbone and gallic acid, which is further metabolized into pyrogallol [77]. On the other hand, the subsequent microbial metabolism of catechin generates phenyl-ɣ-valerolactones—exclusive intermediates of flavan-3-ol degradation—that are further metabolized into hydroxyphenylpropionic acid and hydroxybenzoic acid [77], whose features—better outlined in the phenolic acid section below—include antioxidant, anti-inflammatory, antiviral, antimicrobial and anticancer properties [78]. According to recent in vitro fermentation experiments, flavan-3-ols promote the growth of *Bacteroides*, *Faecalibacterium*, *Parabacteroides* and *Bifidobacterium*, thus suggesting that these genera might be directly involved in the intestinal metabolism of such biomolecules [79].

Phenolic acid metabolites such as vanillic acid, homovanillic acid, hippuric acid and *p*-coumaric acid have also been related to flavan-3-ol metabolism mediated by the gut microbiota [80] and are found methylated, glucuronated, sulfated or nucleated in the bloodstream. These end-products have been reported to possess an antiadhesive effect in *E. coli* urinary infections in a T24 bladder epithelial cell assay [81]. Such results further confirm the reported efficacy of cranberry juice—particularly rich in such biomolecules—in reducing the recurrence of urinary tract infections in women [82].

#### 2.1.6. Anthocyanins

Anthocyanins are natural plant pigments that can be easily found in our daily diet as they are responsible for most of the red, blue or purple color of the fruit. Examples include berries, apples, pears, red-skinned grapes and vegetables, such as radishes (*Raphanus sativus*), purple tomatoes (*Lycopersicon esculentum* ‘*Indigo Rose*’) and red cabbage (*Brassica oleracea var. capitata f. rubra*). Comparable to other classes of flavonoids, anthocyanins—together with their aglycone counterparts called anthocyanidins—have shown antioxidant properties in vitro, with scavenging effects on free radicals [83]. A recent study reported an effective reduction of platelet aggregation induced by arachidonic acid in coincubation with anthocyanins [84], thus supporting the possible supplementation of the latter as an adjunct in the prevention of thrombosis.

In vivo, many anthocyanin compounds have shown neuroprotective and anti-inflammatory effects, comparable to those of acetylsalicylic acid [85,86]. According to several epidemiological and human intervention studies, anthocyanin administration could be a reducing factor for the risk of cardiovascular diseases, due to their anti-thrombotic effects [87,88,89,90]. In addition, anthocyanins can inhibit the replication of viruses such as *Herpes simplex*, human parainfluenza viruses, respiratory syncytial virus, human immunodeficiency virus (HIV), rotaviruses and adenoviruses [91].

Anthocyanins are the glycosylated forms of anthocyanidins, carrying one or several saccharides bound to their scaffold structure. Procyanidins can occur in monomeric as well as in polymeric forms, the latter being responsible for the red, purple and blue colors found in fruits and vegetables [92]. Their backbone structure is the flavylium cation and, depending on the number and position of hydroxyl and methylated groups, various anthocyanidins have been described (Appendix A). The hydration layer constituted by the hydroxyl groups and the polymeric and oligomeric glycone forms contributes to the low bioavailability of anthocyanins in the upper digestive tract. It is therefore very likely that a large numbers of these compounds enter the colon unmodified, where they are processed by the resident microbiota [93]. In fact, only about 1–2% of ingested anthocyanins retain their original structure in the plasma [8]. In the intestine, bacterial β-glucosidases and other enzymes involved in the ring-opening lead to a series of degradation products such as phloroglucinol, vanillic acid and protocatechuic acids [93] known for their antioxidant, antimicrobial and anti-inflammatory properties [94]. Particular emphasis should be given to phloroglucinol, which, in addition to antioxidant and anti-inflammatory properties, was shown to have a relaxing effect on gut smooth muscle during placebo-controlled human trials, with positive effects on patients suffering from irritable bowel syndrome with diarrhea [95] and potential application during esophagogastroduodenoscopy [96]. Other identified microbial end-products include gallic, syringic and *p*-coumaric acids, which have been associated with health-promoting properties [97]; to date, the main bacteria presumably involved have been identified by in vitro microbial cultivations and include *Lactobacillus* spp. (i.e., *L. plantarum*, *Lactobacillus casei* and *L. acidophilus)* and *Bifidobacterium* spp. (i.e., *B. adolescentis*, *Bifidobacterium infantis* and *B. bifidum*) [98], most likely because these genera typically possess β-glucosidases and ring-fission catabolic activities [99].

### 2.2. Stilbenes

Stilbenes constitute a vast group of non-flavonoid plant natural defense biomolecules that act as antifungal compounds, produced especially after a lesion or a fungal infection. Among the over 400 natural stilbenes identified, *trans*-resveratrol (hereinafter simply ‘resveratrol’) is the best known and is mainly found in grape skin. Its properties—besides being a natural phytoalexin—have been extensively studied in the past years, making it the most popular polyphenol. This stilbene and stilbene alkanoic derivatives in general displayed antioxidant and antiproliferative activities on various cancer cell lines such as C2C12 (mouse muscle myoblast) and MCF7 (human breast adenocarcinoma) [100]. In particular, they can promote the activity of antioxidant enzymatic defense systems and increase the efficacy of non-enzymatic compounds, such as glutathione, in scavenging reactive oxygen species [101]. Resveratrol is not the only form of stilbene that has been investigated for its antioxidant capacity. For example, a natural oligomer of resveratrol (i.e., trans-δ-viniferin) is able to scavenge superoxide ions and inhibit lipid peroxidation efficiently. In vitro, stilbenes also act as anti-inflammatory compounds and have been shown to prevent glycation, neurodegeneration and aging [102,103]. Natural extracts rich in stilbenes showed good antimicrobial activities [104] to the point that the scaffold structure of stilbenes has been used to generate highly effective antimicrobials through biotransformation processes [105] (e.g., the so-called “duotap” dimeric stilbene compounds [106]).

The anti-inflammatory and neuroprotective effects of stilbenes, especially resveratrol, have been confirmed in rodent models, with promising results also in terms of cognitive impairment recovery and amyloid plaque reduction [102]. An additional possible use of these compounds in the future is as natural additives to preserve food from oxidation, given their efficacy in these terms [107] and their non-genotoxicity as tested in vivo [108]. Finally, resveratrol has shown promise in treating human respiratory viral infections, given the antiviral activity reported against, to name a few, influenza virus, respiratory syncytial virus and SARS-CoV-2 [109].

The structure of stilbenes (Appendix A) exists in the following two isomeric forms: (E)-stilbene or *trans*-stilbene, and the isomer (Z)-stilbene (*cis*-stilbene), which is less stable. Natural analogues of resveratrol, such as pterostilbenes and viniferins, are higher molecular weight molecules. In particular, pterostilbenes show a higher lipophilicity, thus increasing membrane permeability and improving their bioavailability [110], whereas viniferins are the least lipophilic molecules of this class. Stilbenes can occur as free aglycones or mainly as glycoside forms (called piceids), which require enzymatic cleavage of the saccharide units to be transferred across the intestinal barrier into the circulatory system, and then undergo glucuronidation and other phase II metabolism in the liver. The catalytic action on the glycones is carried out by the gut microbiota as most of the ingested glycosides reach the colon.

To investigate the role of the gut microbiota in the metabolism of stilbenes (and therefore in determining their effectiveness), a recent study tested *Vitis vinifera* extracts by implementing M-SHIME^®^, a validated in vitro model of the intestinal environment [111]. Daily administration of stilbene-rich extracts (up to 1 g/L) led to significant changes in the community metabolism and composition, suggesting a role of the microbiota in the metabolism of such biomolecules. In particular, higher levels of short-chain fatty acids (SCFAs, i.e., the end-products of the fermentation of fibers by the gut microbiota, with a pivotal role in host physiology) and NH_4_^+^—overall considered a clue of wellness of the microbial population—were detected, together with a general increase in Enterobacterales and a decrease in Bacteroidales orders. Furthermore, the authors reported that Gram-negative species were less sensitive to the potential anti-microbial activity of the tested extracts. In another study, fecal samples from different healthy omnivorous donors without a history of antibiotic usage in the previous 3 months were incubated in an in vitro fecal fermentation system to evaluate the microbiota’s ability to digest six stilbenes and stilbenoids [112]. According to the authors’ findings, resveratrol, oxyresveratrol and piceatannol were extensively metabolized by the fecal microbiota, undergoing double bond reduction, dihydroxylation and demethylation, depending on the position of hydroxyl and methyl groups, thus generating various metabolites. For example, resveratrol fermentation resulted in dihydroresveratrol as the only metabolite, as detected by liquid chromatography followed by mass spectrometry (LC/MS). However, it should be noted that a previous study reported two additional compounds, i.e., 3,4-dihydroxy-*trans*-stilbene and 3,4-dihydroxybibenzyl, in almost—but not all—the fecal samples tested, thus suggesting that metabolic processes and end-products are strictly dependent upon individual microbiota composition [113]. The biological properties of the so far identified intermediates and end-products still have to be clearly elucidated, but they might be responsible for the positive effects (e.g., antioxidant, anti-inflammatory and antitumoral properties) observed in some studies after the administration of stilbenes [110,114]. Results obtained from resveratrol and stilbene administration might sometimes be contradictory [115,116]. It is also worth noting that several stilbene-based engineered drugs have been approved by the U.S. Food and Drug Administration agency (FDA) and the European Medicines Agency (EMA) and are effectively in use for estrogen-receptor modulating therapies such as raloxifene (osteoporosis in women), toremifene and tamoxifen (both in use for hormone receptor-positive breast cancer) [117]. Another stilbene derivate, Ramizol, is currently under preclinical investigation for the treatment of *Clostridioides difficile* infections [118,119].

### 2.3. Phenolic Acids

Phenolic acids are other non-flavonoid phenolic compounds, widespread in plants as free or saccharide-conjugated soluble and insoluble forms. Berries, cereals, legumes and oilseeds carry the highest amounts of phenolic acids. Phenolic acids exhibit marked radical scavenging capacity, resulting in a beneficial effect against cancer development, cardiovascular diseases, inflammatory diseases and other disorders [97]. In addition to the antioxidant property, quite common to polyphenols in general, as already discussed, phenolic acids possess other potentially clinically relevant properties. For example, ferulic acid showed marked antithrombotic effects in vitro and in vivo [120]; coumaric acid showed inhibitory effects on lactate dehydrogenase (LDH) release, promoting the recovery of hyperlipidemia steatohepatitis in vivo [121]; gallic acid showed anti-urolithiatic properties (inhibition of urolithiasis crystal formation), thus improving kidney health [122]; vanillic acid significantly inhibited human colorectal cancer growth in a xenograft tumor model, via the inhibition of hypoxia-induced expression of hypoxia-inducible factor (HIF)-1α, thereby inhibiting in a dose-dependent manner the vascular endothelial growth factor (VEGF) and erythropoietin (EPO) proteins, both involved in tumor graft angiogenesis [123].

Phenolic acids have also shown marked antibacterial activity in vitro against several pathogenic strains [124]. Zhang and colleagues [125] reported a synergic effect of coumaric acid and chlorogenic acid with fosfomycin in the treatment of *L. monocytogenes* infections, whilst Tan et al. [126] tested chlorogenic acid with levofloxacin, obtaining positive results on *Klebsiella pneumoniae* infections in vivo. Extracts rich in phenolic acids (e.g., from peanuts) have shown antibacterial effects against Gram-positive bacteria species, such as *Bacillus cereus* and *S. aureus*, and Gram-negative pathogens, such as *Pseudomonas aeruginosa* and *Salmonella enteritidis* [127].

According to their scaffold structure, phenolic acids can be divided mainly into hydroxybenzoic (Appendix A) and hydroxycinnamic (Appendix A) acids. Hydroxycinnamic acids are derived from cinnamic acid and are usually present in foods as esters with glucose units. Hydroxybenzoic acids are derived from benzoic acid and are often found in soluble moieties with various sugars such as glucose and rhamnose. Among hydroxybenzoic acids, the best known are vanillic acid, gallic acid, *p*-hydroxybenzoate and protocatechuic acid, whereas hydroxycinnamic acids are mainly represented by chlorogenic, caffeic, ferulic and coumaric acids.

It has been extensively demonstrated that phenolic acids are metabolized by human phase II metabolism enzymes after absorption in the gastrointestinal tract, undergoing methylation, glucuronidation and sulfation with derivates that may be (more) biologically active [128,129]. Phenolic acids as gut microbiota-derived end-products are produced in most metabolic pathways of phenolic compounds (Appendix A) and have been shown to possess intrinsic anti-inflammatory properties, as well as synergistic anti-inflammatory effects with SCFAs [130]. Hence, the role of the gut microbiota in phenolic compound metabolism, in general, is due to its ability to derive smaller phenolic compounds (i.e., phenolic acids) starting from complex phenolic (a)glycones, otherwise less absorbable and with lower biological activity.

### 2.4. Tannins

Tannins are a class of polyphenolic biomolecules with a high molecular weight (500 Da to 20 kDa) found in most plants, given their pivotal role in protecting from predation and regulating plant growth. The major classes of tannins are hydrolyzable tannins, condensed tannins (also known as non-hydrolyzable tannins) and phlorotannins. Hydrolyzable tannins consist of repeated units of gallic (i.e., gallotannins) or ellagic (i.e., ellagitannins) acid, or other polyhydric alcohols together with a sugar core (Appendix A). Tannic acid, a mixture of digallic acid esters of glucose (Appendix A), is one of the simplest examples of hydrolyzable tannins. Condensed tannins are mainly made of catechins together with anthocyanidin aglycone scaffolds, which explains their alternative name of proanthocyanidines (Appendix A). Finally, phlorotannins are oligomers of phloroglucinol, a compound mainly produced in algae. An additional group of biomolecules that are sometimes included among tannins because of their high molecular weight is polystilbenes, which, as the name suggests, are polymeric structures of stilbenoid scaffolds mainly represented by several viniferins (Appendix A). In vitro tannins have shown marked antioxidant properties linked to the prevention of cardiovascular diseases, cancer and osteoporosis. This potential has also aroused the interest of the food industry, which uses them as preservative agents in food [131]. In addition, tannins have shown promising results in the beef industry, reducing ammonia and methane production in rumen fermentation both in vitro [132] and in vivo [133].

In human studies, the anticancer properties of tannins have been investigated, finding interesting results, particularly for tannic acid, both in cancer prophylaxis and as an adjuvant in cancer therapy [134,135]. Tannic acid has also shown promising antibacterial activity against both Gram-positive and Gram-negative species, such as *S. aureus*, *E. coli*, *Streptococcus pyogenes*, *Enterococcus faecalis*, *P. aeruginosa*, *Yersinia enterocolitica* and *Listeria innocua* [136,137,138]. Furthermore, it has shown antiviral effects against pathogenic viruses such as influenza A virus, Papilloma virus, noroviruses, *Herpes simplex virus* type 1 and 2 and HIV [139].

The pharmacological aspects of tannins have not been investigated as thoroughly as the simpler polyphenols; however, it is clearly known that their intake is not associated with direct adsorption [140], resulting in over 90% of biomolecules entering the colon. Here, the gut microbiota plays a pivotal role in the catabolism of such polymers into their monomers, with the polymerization rate appearing to determine the fate of such compounds during digestion [141]. Gut colonizers as the genus *Akkermansia* and members of the families *Lachnospiraceae* and *Ruminococcaceae* [142], as well as the species *Butyrivibrio* spp., *Gordonibacter urolithifaciens* and *Bifidobacterium pseudocatenulatum* [143], are mostly necessary to ferment non-digestible condensed tannins into bioavailable metabolites—mainly catechins and phenolic acid scaffolds—that can exert systemic pharmacological effects as illustrated in the previous sections. At the same time, fermentation produces useful substrates (e.g., SCFAs and probably other still unknown metabolites) for the microbial counterpart, in a sort of prebiotic effect. In contrast, hydrolyzable tannins are degraded in mild acid conditions, releasing the monomers of gallic or ellagic acid directly in the upper digestive tract, which then undergo adsorption, conjugation with methyl, glucuronic or sulfate groups and finally excretion [144].

## 3. Conclusions

Being at the interface between humans and their environment, the gut microbiota plays a vital role in mediating the biological effects of compounds entering the intestine, including dietary components. Regarding the phenolic nutraceuticals specifically covered in this review, gut microbes are known to be involved in generating a potentially vast diversity of biologically effective end-products, such as phenolic acids, from glycones and polymeric forms. These metabolites, as well as aglycones, are absorbed along the intestine and can therefore exert systemic effects, either in their form or after phase II metabolism, as well as an effect on the microbial counterpart. Eventually, they can still be metabolized by the gut microbiota after intestinal excretion of the conjugated products (Figure 1). It is worth noting that for polyphenol and tannin nutraceuticals, the neologism duplibiotic has recently been proposed [145], precisely to indicate unabsorbed substrates that can modulate the gut microbiota composition through a dual prebiotic and antimicrobial effect that relies on the microbial metabolism of the duplibiotic itself. However, to date, our knowledge of the microbiota-derived metabolites of phenolic nutraceuticals is limited and their effects on human physiology are sometimes not easily explained. Among the main factors involved, there is certainly the great variety of phenolic compounds found in nature, the often-reduced numbers of enrolled patients in clinical trials, and, not least, the inter-individual variability—and even the temporal intra-individual variability—of the gut microbiome. In addition, the studies performed to assess the therapeutic efficacy of phenolic compounds often enrolled healthy subjects, making it more difficult to measure significant changes in biomarkers generally associated with pathological conditions as well as in the microbiota composition, given that dietary supplementation is often performed in the context of an unsupervised diet. Finally, it must be said that studies focusing their attention on the effect of plant-derived nutraceuticals on the gut microbiota often implemented different study designs, including the following: (i) in vitro batch fermentations of phenolic compounds with single bacterial strains, or simple synthetic consortia or complex microbial communities, either derived from animal models (mouse or rat feces, rumen fluid, etc.) or from human donors (mostly feces from healthy donors as mentioned above, and to a lesser extent saliva or other fluids); (ii) intestinal epithelial cell cultures co-incubated with plant extracts or pure phenolic compounds and simplified consortia or donor slurry, to simulate a gut ecosystem; (iii) in vivo interventional studies in animal models or directly in humans. The techniques used to assess the microbial composition and inspect eventual changes were also highly variable, ranging from viable plate counting in selective media to quantitative polymerase chain reaction with group-, genus- or species-specific primers to 16S rRNA gene sequencing (currently the gold standard for microbiota profiling), thus yielding different taxonomic resolution. In this scenario, the use of untargeted metabolomics and high-resolution shotgun metagenomics/metatranscriptomics of the gut microbiome are highly advocated as they could allow the identification of novel end-products and microbial metabolites produced alongside, as well as the enzymatic functions responsible for these conversions and the species that harbor the corresponding genes. Once this knowledge has been achieved, it is very likely that more rational intervention strategies can be implemented based on the individual gut microbiota profile for truly personalized and precision treatments.

## Figures and Tables

**Figure 1 biomolecules-12-00875-f001:**
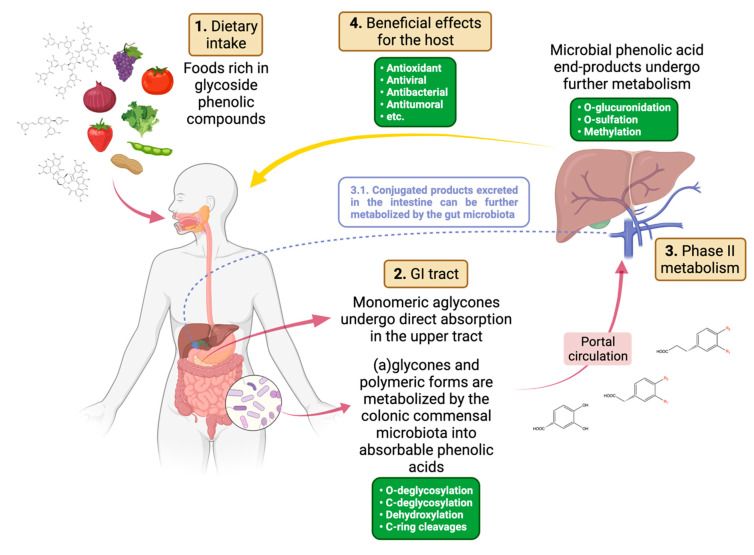
Schematic representation of the journey of phenolic compounds through the human body. Phenolic compounds (i.e., polyphenols and tannins) ingested with food during our daily dietary consumption mostly consist of glycosides or polymerized higher molecular weight moieties, but also include monomeric forms. In the oral cavity and upper digestive tract, a small fraction of such compounds is metabolized by the resident microbiota, releasing small amounts of saccharides and aglycones. Whether the monomeric aglycones come from this first step of degradation or directly from ingested food, they undergo absorption through the intestinal epithelium. Most of the phenolic moieties—the fraction of which depends on their structure as well as the food matrix they were embedded into—reach the colon unchanged and come into contact with the trillions of commensal microorganisms. Here, deglycosylation reactions release saccharides from the sugar moieties, then dehydroxylation and phenolic ring-targeted metabolism break down the complex phenolic structures into simple and absorbable phenolic acids, among which protocatechuic acid, 3-(3,4-dihydroxyphenyl) acetic acid and 3-(3,4-dihydroxyphenyl) propionic acid. Such metabolic processes provide support for microbial growth, contributing to the production of complex microbial metabolites, still largely unknown. The phenolic end-products are absorbed through the epithelium and—as for the simple monomeric aglycones absorbed in the upper intestine—reach the liver via the portal circulation, where they undergo phase II metabolism reactions such as glucuronidation, sulfation and methylation. The resulting glucuronate, sulfate and methyl compounds are found in the bloodstream and are most likely responsible for the beneficial effects reported, including antitumoral, antioxidant, antibacterial and antiviral effects. Eventually, they can still be metabolized by the gut microbiota after intestinal excretion of the conjugated products. GI, gastrointestinal. Image created in BioRender.com.

## Data Availability

Not applicable.

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
