# Peer review of "Polyphenol and Tannin Nutraceuticals and Their Metabolites: How the Human Gut Microbiota Influences Their Properties"

_biomolecules, 2022, doi:10.3390/biom12070875_

Round 1

Reviewer 1 Report

Dear Editor,

Thank you for the opportunity to review the paper entitled "Polyphenol and tannin nutraceuticals and their metabolites: how the human gut microbiota influences their properties". The paper is well-structured and written in the scientific manner. The used literature is appropriate and up to dated. Although the extensive researches have been conducted related to polyphenol and tannin nutraceuticals this review systematically covers the main results in the available literature in this area. I would recommend the acceptance of this paper after minor revision. Namely, in order to cover all topics and concepts, please discus further about the following:

-          Concept of polyphenol duplibiotic effect (this part is completely missing and this concept has been introduced last year)

-          Also, I would recommend the discussion about the experimental set-ups in testing the influence of polyphenol on the gut microbiota (the experiments with individual cells, simplified consortium, in vitro colonic fermentations and in vivo studies).

In my opinion the quality of the paper would be significantly improved by the addition of discussion of the abovementioned concepts.

Author Response

Thank you for the opportunity to review the paper entitled "Polyphenol and tannin nutraceuticals and their metabolites: how the human gut microbiota influences their properties". The paper is well-structured and written in the scientific manner. The used literature is appropriate and up to dated. Although the extensive researches have been conducted related to polyphenol and tannin nutraceuticals this review systematically covers the main results in the available literature in this area. I would recommend the acceptance of this paper after minor revision.

Answer: Thank you for the appreciation and for the time spent reviewing our manuscript!

Namely, in order to cover all topics and concepts, please discus further about the following:

- Concept of polyphenol duplibiotic effect (this part is completely missing, and this concept has been introduced last year)

Answer: We thank the Reviewer for the valuable suggestion. We agree that the concept fits perfectly the scope of this review and sounds brilliant. Accordingly, we added the concept to the Conclusions section of the manuscript (please, see lines 537-541).

-  Also, I would recommend the discussion about the experimental set-ups in testing the influence of polyphenol on the gut microbiota (the experiments with individual cells, simplified consortium, in vitro colonic fermentations and in vivo studies).

In my opinion the quality of the paper would be significantly improved by the addition of discussion of the abovementioned concepts.

Answer: We greatly appreciate the Reviewer’s recommendation, and we agree that clarification of the experimental set-ups would add value to the manuscript. Indeed, we expanded the discussion concerning experimental design and generally adopted techniques that have been cited throughout the manuscript (lines 551-564).

Reviewer 2 Report

The topic of the review is very interesting. However there are different points that authors should address:

Lines 26-28: Probiotics are not always derived from food. In fact, many of the most used probiotic strains are from human origin. Therefore, I suggest authors to re-write the paragraph, since it does not truly represent the reality.

Line 34: In my opinion, authors could state that there are different classifications for polyphenols, and to indicate which of these classifications are they using. Please, re-check whether the reference 3 classifies polyphenols as it is stated in lines 41-42.

Line 43: phenolic acids can be classified not only in the 2 groups indicated. Authors may find useful the following link: http://phenol-explorer.eu

Line 62: I think the objective of the study (written in lines 28-34) should be moved here, at the end of the introduction section.

In addition, I think authors should include a paragraph where the literature searching strategy is presented. This is very important, for whichever kind of review (not only for systematic reviews).

Lines 610-611: in vitro fermentation is a method to study microbial fermentation. 16SrRNA gene sequencing is a method to study microbial populations, and it could be used to analyse samples from in vitro fermentations of from human feces, for example. I think authors are confusing study designs with methodology for microbiota analysis, so this sentence does not make any sense. Please check this.

To be taken into account throughout the manuscript:

-       In vitro and in vivo studies should be described in more detail (doses administrated, type of cells, which and how many animals, administration duration, type of population, etc…)

-       Although in the literature the terms “microbiota” and “microbiome” are many times wrongly used indistinctly, they mean different things. Please, be aware of this and write the proper term when using them.

-       When referring to bacteria, please specify each time when are you talking about families, genera, species or strains.

-       In my opinion the manuscript is too long. Since chemical structures are already available in the literature, I think authors should focus more on the impact of gut microbiota on polyphenol and tannin properties and less in the structures.

Author Response

The topic of the review is very interesting. However there are different points that authors should address:

Lines 26-28: Probiotics are not always derived from food. In fact, many of the most used probiotic strains are from human origin. Therefore, I suggest authors to re-write the paragraph, since it does not truly represent the reality.

Answer: We thank the Reviewer for the rectification and apologize for the inaccuracy. Probiotics are indeed derived from other different sources. However, given that probiotics are not relevant to the purposes of this manuscript, we opted to remove the word from the revised version (line 27), to improve the focus on the discussed topic.

Line 34: In my opinion, authors could state that there are different classifications for polyphenols, and to indicate which of these classifications are they using. Please, re-check whether the reference 3 classifies polyphenols as it is stated in lines 41-42.

Line 43: phenolic acids can be classified not only in the 2 groups indicated. Authors may find useful the following link: http://phenol-explorer.eu

Answer: The esteemed Reviewer is totally right, and we thank her/him for the provided website that was extremely helpful in checking the classification we adopted in the manuscript. Reference 3 was indeed at odds with the mentioned classification and has been replaced by a reference pointing to the phenol-explorer database (line 36). In particular, as noted by the Reviewer, our manuscript focused on most but not all of the classes reported in such a database. In an attempt to improve clarity, in the revised version of our manuscript, we specified that the mentioned classes and families did not consist in all existing ones but rather in a representative set (lines 60-61). As regards the polyphenol groups not included, it must be said that for example the properties of the derivates of hydroxyphenyl acetic acids are discussed through the manuscript as they are found as end-products for some of the presented cases of phenolic compound degradation (lines 130-133 and 162). We have chosen not to add new paragraphs on the groups not considered, also in the light of the Reviewer’s comment on the excessive length of the text, but if the Reviewer deems it appropriate, we will be more than willing to include new parts for a more comprehensive discussion.

Line 62: I think the objective of the study (written in lines 28-34) should be moved here, at the end of the introduction section.

Answer: We thank the Reviewer for the suggestion. We applied the edit to the manuscript and made the objective of the study clearer (please, see lines 59-65 and also the comment above).

In addition, I think authors should include a paragraph where the literature searching strategy is presented. This is very important, for whichever kind of review (not only for systematic reviews).

Answer: According to the Reviewer’s suggestion, we added an explanatory sentence at the end of the Introduction (lines 65-69).

Lines 610-611: in vitro fermentation is a method to study microbial fermentation. 16SrRNA gene sequencing is a method to study microbial populations, and it could be used to analyse samples from in vitro fermentations of from human feces, for example. I think authors are confusing study designs with methodology for microbiota analysis, so this sentence does not make any sense. Please check this.

Answer: We totally agree with the Reviewer, the sentence was not clear. We re-wrote the paragraph, also in the light of Reviewer #1’s comments, clarifying on the one hand the most used study designs and, on the other, the methods for profiling the microbial populations coupled to such designs (lines 551-564).

To be taken into account throughout the manuscript:

-       In vitro and in vivo studies should be described in more detail (doses administrated, type of cells, which and how many animals, administration duration, type of population, etc…)

Answer: We thank the Reviewer for the valuable suggestion. In the revised version of the manuscript, we have included a new table, Supplementary Table 1, which summarizes for each study considered the compound used, the dose administered, the duration of administration, and the study design. 

-       Although in the literature the terms “microbiota” and “microbiome” are many times wrongly used indistinctly, they mean different things. Please, be aware of this and write the proper term when using them.

Answer: The Reviewer’s comment is very accurate and we apologize for the sometimes improper use of the two terms. We have fixed the manuscript accordingly, using the term “microbiota” in lines 112, 409, 514, 529 and 570 (where we refer specifically to the taxonomic composition), and the term “microbiome” in lines 547 and 566, where the sentence refers not only to microbes, but also to the metabolic and functional properties of the ecosystem itself. 

-       When referring to bacteria, please specify each time when are you talking about families, genera, species or strains.

Answer: We thank the Reviewer for the suggestion. Corrections have been made throughout the manuscript (lines 115, 116, 128, 266, 315, 461, 506, 516, 517).

-       In my opinion the manuscript is too long. Since chemical structures are already available in the literature, I think authors should focus more on the impact of gut microbiota on polyphenol and tannin properties and less in the structures.

Answer: We thank the Reviewer once again for the suggestion. Following her/his advice, we have moved the previous Figures 1, 2 and 3 into the Supplementary Material as Supplementary Figures 1, 2 and 3. We think that getting rid of those figures completely would be unfavorable for the manuscript, as the graphical representation could help less familiar readers to understand the wide diversity of polyphenol scaffolds and glycone forms. In any case, if the Reviewer still finds the figures useless, we will be more than willing to remove them altogether. In addition, to further shorten the manuscript and centralize the focus on the impact of the gut microbiota, we removed several sentenced on the chemical structure of the various polyphenol classes (lines 101, 154, 167, 200, 232, 235, 300-302, 343, 394).

Round 2

Reviewer 2 Report

Thank you for taking in consideration my comments.